# The Polymorphism of Metabolic and Immune Mechanisms Controlling Genes in Type 2 Diabetes Mellitus

**DOI:** 10.3390/genes16091116

**Published:** 2025-09-20

**Authors:** Iuliana Shramko, Elizaveta Ageeva, Anatolii Kubishkin, Tatyana Makalish, Cyrill Tarimov, Dmitry Bondar’

**Affiliations:** 1Department of General and Clinical Pathophysiology, Order of the Red Banner of Labor S.I. Georgievsky Medical Institute of the Federal State Autonomous Educational Institution of Higher Education, V. I. Vernadsky Crimean Federal University, Lenin boulevard, 5/7, 295000 Simferopol, Russia; kubyshkin_av@mail.ru (A.K.); kirito.k@yandex.ru (C.T.); 2Department of Medical Biology, Order of the Red Banner of Labor S.I. Georgievsky Medical Institute of the Federal State Autonomous Educational Institution of Higher Education, V. I. Vernadsky Crimean Federal University, Lenin boulevard, 5/7, 295000 Simferopol, Russia; ageevaeliz@rambler.ru (E.A.); office@ma.cfuv.ru (D.B.); 3Center for the Collective Use of Scientific Equipment “Molecular Biology”, V. I. Vernadsky Crimean Federal University, Lenin boulevard, 5/7, 295000 Simferopol, Russia; makalisht@mail.ru

**Keywords:** metabolic syndrome, interleukin-6, genetic polymorphisms

## Abstract

**Background/Objectives**: Most genes involved in the pathogenesis of Metabolic Syndrome (MS) and Type 2 Diabetes Mellitus (T2DM) are regulated by peroxisome proliferator-activated receptors (PPARs), which modulate the production of pro-inflammatory cytokines, with interleukin-6 (IL-6) playing a crucial role. The associations of single-nucleotide polymorphisms (SNPs) with MS and T2DM remain uncertain across populations. Therefore, we aimed to investigate the associations of PPAR-related SNPs in *IL-6*, *LEP*, *ADIPOQ*, *ADIPOR1*, and *ADIPOR2* with MS and T2DM clinical features. **Methods**: Polymorphism analysis of *IL-6*, *LEP*, *ADIPOQ*, *ADIPOR1*, and *ADIPOR2* genes was performed on isolated DNA from individuals diagnosed with T2DM and from healthy controls using real-time polymerase chain reaction (qPCR). **Results**: The *IL-6-174G/C* polymorphism shows that the CC genotype is associated with higher MS risk, whereas the GG genotype appears protective against metabolic disturbances. When the *IL6* CC genotype is combined with *ADIPOR2* GA or *ADIPOR2* 219 A/T, hyperglycemia is 1.3 times more frequent than with other *IL6/ADIPOR2* genotype combinations. **Conclusions**: To develop informative genetic risk scores, future studies should include additional polymorphisms in key immune–metabolic pathway genes, such as *AP-1*, *NF-κB*, and *FFAs.*

## 1. Introduction

The metabolic and immune systems play a crucial role in maintaining homeostasis and normal physiological functions. These systems have co-evolved and are tightly interdependent. A range of hormones, cytokines, signaling proteins, transcription factors, and bioactive lipids can function both metabolically and immunologically [1]. Among metabolic diseases, metabolic syndrome (M) and type 2 diabetes mellitus (T2DM) are highly prevalent worldwide. According to the International Diabetes Federation (IDF), approximately 537 million patients with T2DM were registered globally in 2021, with a projected increase to 783 million by the year 2045 [2]. The interaction between environmental factors, including drugs and chemicals [3], and genetic components plays a significant role in the etiology and pathogenesis of T2DM and MS [4].

A high-calorie diet currently is proved to be the primary stimulus that triggers hyperinsulinemia, leading to an excessive accumulation of lipids within adipose cells. The hypertrophied adipocytes produce significant concentrations of cytokines, which, in combination with impaired angiogenesis in adipose tissue, leads to hypoxia. Hypoxia acts as a trigger for adipocytes’ death via ischemic necrosis and apoptosis. This process stimulates the recruitment of M1-phenotype macrophages and the polarization of resident adipose tissue macrophages into M1-phenotype macrophages [5]. Inflammatory signals produced by M1-macrophages further contribute to the development of insulin resistance [6], while inhibition of these inflammatory signals disrupts this process [7]. Most obesity-induced cytokines decrease tissue sensitivity to insulin by activating self-perpetuating cycles involving tumor necrosis factor-alpha (TNFα) and nuclear factor kappa B (NF-κB) [8]. Due to cooperation between enlarged adipocytes and macrophages (M1), TNFα interacts with its own receptor on the adipocyte membrane, stimulating the hydrolysis of neutral fats into free fatty acids (FFAs). Similar to lipopolysaccharides (LPS), FFAs activate toll-like receptor 4 (TLR4) and NF-κB, triggering the expression of several pro-inflammatory genes, including TNFα. Thus, a vicious circle is formed under conditions of a high-calorie diet (a source of FFA) and hypertrophy of adipocytes (a source of both TNFα in hypoxia and free radicals—in oxidative stress), resulting the aggravation of insulin resistance [9].

The activation of immune-metabolic cascades is associated with changes in the expression of key genes regulating adipocytes’ differentiation, glucose transport and insulin sensitivity, lipid metabolism, oxidative stress and inflammation. Most of the genes mentioned are under the transcriptional control of PPAR (Peroxisome proliferator-activated receptors) [10]. PPAR family proteins suppress the activity of other nuclear transcription factors (NTF), in particular NF-kB and AP-1 (activating protein type 1), carrying out the mechanism of negative regulation of inflammation by the transrepressional transcription of AP-1 and NF-kB target genes [11]. It was found that activation of PPAR-γ stimulates adiponectin secretion, while production of leptin, resistin, cytokines (TNF-α, IL-6), plasminogen activator inhibitor-1 (PAI-1), and cortisol in adipocytes is suppressed. PPAR-α ligands (including FFA [12]) inhibit activation of the *IL-6* gene promoter [12,13].

Several studies have reported associations between SNPs and MS/T2DM clinical features. Our previous work [14] identified associations between certain *LEP* and *ADIPOQ* genotypes (and their receptors) and risks of hypertension, hyperglycemia, and other MS/T2DM manifestations. Banerjee and Saxena [15] reported that the IL6 promoter SNP-174G/C (*rs1800795*) modulates transcription in response to inflammatory stimuli (e.g., LPS, IL-1β). Studies by Tiis and Osipova [16] have stated that -174G/C polymorphisms of the *IL6* gene are associated with an increased risk of T2DM among carriers of certain genotypes, while others have not established valid relationships [17]. Therefore, it is important to study gene–gene interactions in MS/T2DM pathogenesis. The intergenic models proposed by Isakova et al. [18] include ADIPOQ as a contributor to insulin sensitivity, but research on interactions with other PPAR-γ–associated genes remains limited.

In connection with the above, the aim of this study was to investigate the associations between common genotypes and genotype combinations for *IL6*, *leptin*, *ADIPOQ*, *ADIPOR1*, and *ADIPOR2* genes with MS components in individuals with T2DM.

## 2. Materials and Methods

### 2.1. Patient Selection

A single-center, case–control study, performed simultaneously on samples of patients with T2DM and healthy residents of the Republic of Crimea was performed. Ninety-four patients (54 females and 40 males) treated at Semashko Republican Hospital, Simferopol, and 100 healthy individuals (59 females and 41 males) as a control group were involved in the study (Table 1). The study was performed according to the guidelines of the Declaration of Helsinki 1975 (revised in 2013) and approved by the V.I. Vernadsky Crimean Federal University Ethics Committee (Protocol No. 8, 17 January 2018). Informed consent was obtained from all subjects involved in the study.

### 2.2. Inclusion and Exclusion Criteria in the Study


*Inclusion Criteria for Participants*


Male or female participants aged 52 to 70 years.Participants with a confirmed diagnosis of T2DM by the International Diabetes Federation (IDF) 2005: abdominal obesity and two additional components: increased triglyceride (TG) levels > 1.7 mmol/L or medication to lower TG levels; low-density lipoprotein (LDL) levels < 1.03 mmol/L in men and <1.29 mmol/L in women, or specific medication; target fasting plasma glucose (FPG) > 5.6 mmol/L, or a prior diagnosis of T2DM; hypertension (blood pressure ≥130/85 mmHg), or antihypertensive medication. Body mass index (BMI) > 30 kg/m^2^.
Willingness to participate in the study and ability to provide informed consent.
*Exclusion Criteria*
Male or female participants aged less than 52 or greater than 70 years of age.A severe course of T2DM, with a target glycated hemoglobin (HbA1c) level > 7.0% and FPG > 7.0 mmol/L two hours after meals (or >9.0 mmol/L) [21].Chronic renal disease, heart failure, liver dysfunction, or malignancy.Inability or unwillingness to participate in the study or sign an informed consent form.
*Inclusion criteria for control subjects*:
Normoglycemic individuals, male or female, with no history of glucose intolerance or family history of diabetes.Aged 52 to 70 years.HbA1c < 6.4% or normal oral glucose tolerance test.BMI < 30 kg/m^2^.Willingness to participate voluntarily in the study and ability to sign an informed consent form.


### 2.3. Biochemical Measurements

Fasting levels of cholesterol, glucose and HbA1c were measured using standard methods

### 2.4. Physical Examination

BMI (body mass index), which is used to diagnose overweight and obesity and to assess its severity, is calculated by dividing body weight (in kilograms) by the square of height (in meters), and reported as kg/m^2^. Hypertension is defined as a diastolic blood pressure ≥90 mm Hg and/or a systolic blood pressure ≥ 140 mm Hg.

### 2.5. PCR Technique

DNA was extracted using phenol–chloroform, and polymorphisms were detected by real-time polymerase chain reaction (qPCR) on a CFX96 thermal cycler (Hercules, CA, USA). Allele frequencies for *IL-6-174G/C (rs1800795*) were assessed by allele-specific qPCR; the remaining SNPs (*LEP-2548 A/G*, *ADIPOQ +45 T/G*, +*276 G/T*, *ADIPOR1-106 T/C*, +*102 T/G*, *ADIPOR2* +*219 A/T*, +*795 G/A*, and *ADIPOR2-2548 G/A*) were assayed by qPCR using Syntol kits (Moscow, Russia).

The research was conducted at the Center for Shared Use of Scientific Equipment (Molecular Biology), V.I. Vernadsky Crimean Federal University.

### 2.6. Statistical Analysis

The data obtained were analyzed using the Statistica 8.0 software package. Qualitative data are summarized by medians (Me) and interquartile ranges (quartiles, Q1–Q3) and relative frequencies (percentages, %). Between-group differences in HbA1c, plasma glucose, BMI, and arterial blood pressure were assessed with the Mann–Whitney U test. A critical level of significance of *p* < 0.05 was adopted. The statistical power of the study was ≈0.8.

Allele frequencies were compared using the chi-square test (χ^2^) with Yates’ continuity correction. Odds ratios (ORs) and 95% confidence intervals (CIs) were calculated.

## 3. Results

The results of the study demonstrated that among patients with T2DM, not only the ratio of studied genes’ allelic combinations but also the average value of the MS components in the blood are varied. In the most common TT genotype (51.1%) of the *ADIPOQ* gene, +45 T/G polymorphism were common with both the lowest values of cholesterol and diastolic pressure compared to the GT and GG variants. Carriers of the GG genotype of the named polymorphism demonstrated higher levels of fasting glucose, glycated hemoglobin, diastolic blood pressure, and IL-6 concentrations (Table 2).

Carriers of the genotype GT of the *ADIPOQ* gene’s +276 G/T polymorphism demonstrated a higher level of diastolic blood pressure (Table 3).

The TG allele combination of the *ADIPOR1* gene’s polymorphism +102 T/G was prevalent among patients with T2DM (50.0%). When examining the indicators among carriers of this genotype, the concentration of HbA1c was found to be significantly higher (*p* < 0.05), whereas the level of IL-6 was lower compared to carriers of other genotypes of this polymorphism. The frequency of a particular genotype (–106T/C *ADIPOR1*) in another polymorphism was also present in 50% of the patients. Individuals with this genotype had increased BMI and IL-6 concentrations compared to those with other alleles. The carriers of other variants of this gene also were found with higher systolic and diastolic blood pressure, as well as with elevated hemoglobin HbA1c and cholesterol levels (Table 2 and Table 3).

In the carriers of the TT-genotype of *ADIPOR1*-106T/C polymorphism the highest systolic and diastolic blood pressures were diagnosed (Table 3). In contrast, carriers of the CC-genotype had significantly increased concentrations of both HbA1c and cholesterol, compared to carriers of other genotypes (TT and CT-106T/C *ADIPOR1*) (*p* < 0.05) (Table 2).

An analysis of the *ADIPOR2* gene +219 A/T polymorphism allelic variations’ prevalence revealed a roughly equal distribution among patients with T2DM—the genotypes AA and AT were observed with frequencies of 37.6% and 39.8%, respectively. However, patients with these genotypes were found with higher levels of HbA1c, blood glucose, and IL-6 (Table 2). An analysis of the association between the +795 G/A polymorphism of the *ADIPOR2* gene and clinical features in patients with T2DM revealed that individuals carrying the GG genotype demonstrated physiological meanings of total cholesterol and diastolic blood pressure (*p* < 0.05), compared to those with other allele combinations. Conversely, individuals carrying the GA genotype demonstrated a suggestive rise in BMI and IL-6 concentration, compared to the other genotypes in this polymorphism (*p* < 0.05).

Analysis of the symptoms’ severity in patients with T2DM depending on the genotype of the *LEP* + 2548 G/A polymorphism indicated the following differences. Carriers of the GG genotype showed a significant elevation in systolic blood pressure (*p* < 0.05). The highest concentration of IL-6 was observed in individuals with the GA genotype, while patients with the AA genotype had lower cholesterol concentrations and higher systolic blood pressure.

It was found that the distribution of the *IL6* gene’s -174G/C polymorphism was associated with higher levels of HbA1c and hyperglycemia prevalence in individuals with the CC genotype. In individuals with the GG genotype, the most notable increases were seen in systolic blood pressure and the concentration of IL-6 (Table 2 and Table 3).

An analysis of the various combinations of polymorphisms revealed that the presence of the GC genotype of the *IL6* gene’s -174G/C polymorphism was more prevalent in combination with the GG genotype for polymorphism +795 G/A in the *ADIPOR2* gene compared to other combinations of these two genes.

When the CC genotype (-174G/C *IL6*) is combined with the GA + 219A/T genotype in the *ADIPOR2* gene, hyperglycemia is 1.3 times more frequent than with other IL6/ADIPOR2 genotype combinations.

## 4. Discussion

Changes in the concentrations of PPAR-related adipokines (adiponectin and leptin) and cytokines (IL-6) may be modified depending on various factors that affect their expression levels [22,23]. These factors may include genetic components, such as the existence of specific high- or low-expression variants of certain genotypes. For instance, a drop in adiponectin levels is associated with an increase in IL-6 expression in lymphocytes, indicating an inadequate anti-inflammatory response of adiponectin. In obese patients, there is a tendency for metabolic inflammation, as measured by levels of IL-6, to increase as insulin sensitivity worsens. IL-6, among pro-inflammatory cytokines, plays a key role in activating the immune response by promoting lymphocyte differentiation and the synthesis of acute-phase proteins in the liver. It also affects hormonal changes associated with endocrine disorders [24].

The analysis of our results demonstrated elevated concentration of IL-6 in patients with MS, which reflects the general dynamics of IL-6 changes in the disease and consistent with the research data [25,26,27,28].

Based on the hypothesis that gene polymorphism affects the level of IL-6, we analyzed the results of genotyping for the -174G/C (*rs1800795*) variant of the *IL6* gene in MS patients. We found that MS patients predominantly have highly proliferative GG/GC -174G/C genotypes of the *IL6* gene (*rs1800795*). Similar findings have been reported by other researchers in their studies. For example, Sumerkina et al. [29] found higher expression of *IL-6* gene in MS in lymphocytes isolated from peripheral blood when incubated in vitro with adiponectin, a combination of adiponectin and leptin, or in the presence of saline. However, no changes in *IL-6* gene expression were observed under the influence of a high concentration of leptin, which may indicate leptin resistance.

Due to the multifactorial nature of MS and T2DM, it is worthwhile to investigate the association between candidate genes and the development of obesity as well as a cluster of metabolic parameters. Recent research has paid sufficient attention to individual SNPs and their interactions in the development of these conditions [30,31,32]. However, it has become apparent that simply identifying candidate genes is insufficient to predict the risk of MS and T2DM development, as well as their individual, socially significant complications such as hypertension and associated cardiovascular pathology.

To search for genetic variations that contribute to the development of specific diseases, researchers conduct studies using a genome-wide association study (GWAS) [33]. A significant number of these GWASs rely on the genetic data from patients stored in biobanks. Based primarily on the results from GWAS, genetic risk scores (GRS) are created, which include variations in nucleotide sequences that collectively pose the greatest risk for the development of a given pathology. In the combined assessment of T2DM risk using genetic data and traditional risk factors, GRS has proven to be the most accurate approach [34]. GRS represents a promising tool for the development of personalized approaches to patient care. It allows not only to predict an individual’s risk of developing a specific condition, but also to determine how effective preventive measures may be in different individuals [35].

However, a large volume of data makes the diagnosis challenging and requires the selection from a vast array of genetic studies. Specific genotypes or alleles from a certain number of genes can be used for predictive testing purposes.

We analyzed the severity of manifestations of MS components in carriers of different genotypes from the panel of genes we examined. Based on an analysis of several statistically significant changes, we were able to identify the most significant risk and/or protective genotypes for early screening of the population for each individual.

Thus, carriers of the GG genotype for the +795 G/A polymorphism in the *ADIPOR2* gene had lower diastolic blood pressure compared to carriers with other allelic combinations for this polymorphism. Conversely, carriers of the GG genotype at the +2548 G/A site in the *LEP* gene exhibited the highest systolic blood pressure levels (see Table 2 and Table 3).

The highest concentration of IL-6 was observed in heterozygotes for the +795 G/A *ADIPOR2* polymorphism. In addition to this, we confirmed the association between CC genotype of the C-174G variant of IL-6 gene and hyperglycemia as one of the key symptoms of both MS and T2DM. It should be noted, however, that there are debates about the role of IL-6 in the main genetic risk scales for MS and T2DM [36]. The absence of significant associations between the CC genotype of the C-174G variant of *IL-6* gene and other studied genotypes or polymorphisms suggests by the limitations of our study due to the small sample size and single population. Additional polymorphisms from key genes in immune–metabolic pathways should be included in future studies of wider populations with longitudinal design to enhance the existing GRS with personalized recommendations.

## 5. Conclusions

As a result of our research, we have found that the presence of the C-174G variant of the *IL-6* gene is associated with an increased risk of MS when the CC genotype is present. However, the GG genotype of the C-174G polymorphism of *IL-6* has been identified as a protective factor and is not linked to an increased risk for the disease. In order to further understand the molecular genetic mechanisms involved in the regulation of both normal and pathological conditions, it is important to investigate the expression and activity of additional genes’ polymorphisms included in the immune–metabolic pathway (e.g., AP-1, NF-κB, FFAs), as well as the expression and activity of signaling cascades. To this end, it is crucial to build population-wide databases of gene variations and develop predictive screening algorithms for MS/T2DM to identify individuals at the early stage of insulin resistance.

## Figures and Tables

**Table 1 genes-16-01116-t001:** Characteristics of the whole study cohort.

Parameters	Standard Recommended Values [19]	Control Group		T2DM Patients	
		Me	Q1–Q3	Me	Q1–Q3
Age(years)	-	61	52–70	61	52–70
HbA1c (%)	7	4.8	4.1–6.0	8.45	7.15–9.95
Fasting plasma glucose level (mmol/L)	5.6–6.9	5.2	3.6–5.8	9.2 *	6.1–11.1
Cholesterol (mol/L)	4.9	4.6	3.6–6.2	5.1	4.6–7.3
IL-6 (pcmol/L) [20]	5.2	4.22	2.45–7.78	8.1	6.5-8.5
BMI (kg/m^2^)	25	24.6	21.4–28.9	33.9 *	26.0–38.7
Systolic blood pressure (mmHg)	120	110	90–118	130.0 *	110–140
Diastolic blood pressure (mmHg)	80	72	65–80	85.0	80.0–90.0

Notes. Me: median; Q1–Q3: the first and third quartiles. * *p* value < 0.05 as compared to control. *—*p*-value < 0.05 as compared to control; abbreviations are given in text below.

**Table 2 genes-16-01116-t002:** Blood levels of MS components and IL-6 levels in the group of healthy individuals and patients with DM2, stratified by allelic combinations of *IL-6*, *LEP*, *ADIPOQ*, and *ADIPOR* polymorphisms (Me (Q1–Q3)).

		HbA1c (%)	Glucose (mol/L)	Cholesterol (mol/L)	IL-6, pcmol/l
Control group		4.8 (4.1–6.0)	5.2 (3.6–5.8)	4.6 (3.6–6.2)	4.22 (2.45-7.78)
**T2DM patients depending on allelic combinations of gene polymorphism**					
**+45 T/G** **(rs2241766)** ** *ADIPOQ* **	TT	8.65 (6.8–10.2)	9.4 (6.3–11.7)	4.9 (4.2–7.9)	6.3 (4.0–7.1)
	GT	8.6 (6.8–10.3)	8.8 (6.9–11.0)	5.7 (4.8–7.3)	5.3 (3.9–6.2)
	GG	8.9 (8.4–9.5)	10.4 (6.1–12.0)	5.4 (4.8–6.7)	7.7 (5.9–7.3)
**276 G/T** **(rs1501299)** ** *ADIPOQ* **	GG	8.5 (6.5–9.9)	**10.2 (6.3–11.6) ***	5.3 (4.6–7.3)	8.8 (6.9–9.0)
	GT	8.6 (7.5–10.3)	8.8 (6.1–10.7)	5.6 (4.8–7.2)	6.9 (3.3–7.0)
	TT	9.0	8.4	6.6	6.0 (4.8–6.3)
** *+* ** **102 T/G** **(rs2275737)** ** *ADIPOR1* **	TT	6.6 (6.1–8.4)	6.6 (5.1–9.1)	5.8 (4.7–6.8)	7.9 (2.3–7.0)
	TG	**9.0 (7.2–10.5) ***	9.6 (7.9–11.8)	5.6 (4.9–7.3)	5.9 (2.1–6.0)
	GG	8.8 (8.3–9.9)	9.8 (6.9–11.0)	5.1 (4.6–6.2)	6.8 (5.1–7.0)
**– 106T/C** **(rs2275738)** ** *ADIPOR1* **	TT	8.7 (7.6–9.9)	**10.0 (6.2–11.3) ***	5.3 (4.8–6.7)	8.0 (4.1–8.6)
	CT	8.3 (6.8–10.0)	9.2 (7.3–11.2)	5.4 (4.4–7.8)	8.9 (5.7–8.9)
	CC	9.0 (6.2–10.2)	8.8 (6.3–10.7)	**6.2 (4.9–7.0) ***	7.7 (5.3–8.1)
**+219 A/T** **(rs11061971)** ** *ADIPOR2* **	AA	8.8 (7.6–9.9)	9.75 (7.9–11.3)	5.7 (4.8–6.7)	7.1 (4.9–7.7)
	AT	8.7 (6.5–10.5)	9.8 (5.8–11.7)	5.2 (4.6–6.4)	7.2 (6.1–8.2)
	TT	7.9 (6.7–9.0)	8.3 (6.9–10.0)	5.4 (4.9–6.5)	5.1 (3.2–4.9)
**+795 G/A** **(rs16928751)** ** *ADIPOR2* **	GG	8.4 (6.8–10.1)	8.8 (6.1–11.3)	**4.9 (4.4–7.3) ***	9.1 (7.6–9.1)
	GA	8.3 (6.2–10.5)	7.9 (6.2–11.4)	5.9 (5.4–10.5)	**10.3 (7.2–10.1) ***
	AA	7.8 (6.0–8.5)	9.5 (8.7–10.0)	5.5 (4.3–6.3)	9.1 (8.1–10.0)
**+2548 G/A** **(rs7799039)** ** *LEP* **	GG	**8.4 (6.8–10.0) ***	**9.2 (7.8–10.6) ***	5.2 (4.2–6.5)	8.7 (6.1–8.0)
	GA	**8.8 (6.7–10.3) ***	**9.4 (6.9–11.4) ***	5.6 (4.8–7.2)	7.0 (6.1–7.8)
	AA	**8.4 (6.8–9.5) ***	8.3 (6.1–11.2)	4.9 (4.8–6.8)	8.0 (5.1–8.0)
**–174G/C** **(rs1800795)** ** *IL6* **	CC	9.4 (6.7–9.5)	9.5 (6.9–11.3)	5.4 (5.0–5.7)	9.33 (4.56–18.2)
	GG	8.4 (7.2–10.1)	8.8 (6.2–10.3)	5.4 (4.6–6.3)	10.8 (4.6–17.5)

Notes. Me: median; Q1–Q3: the first and third quartiles; HbA1c, glycosylated hemoglobin. * *p* value < 0.05 as compared to control (highlighted in bold).

**Table 3 genes-16-01116-t003:** Blood pressure and body mass index in the group of healthy individuals and patients with DM2, stratified by allelic combinations of *IL-6*, *LEP*, *ADIPOQ*, and *ADIPOR* polymorphisms (Me (Q1–Q3).

		BP Systolic (mmHg)	BP Diastolic (mmHg)	BMI (kg/m^2^)
Control group		110 (90–118)	72 (65–80)	24.6 (21.4–28.9)
**T2DM patients depending on allelic combinations of gene polymorphism**				
**+45 T/G** **(rs2241766)** ** *ADIPOQ* **	TT	130 (110–140)	83 (80–90)	33.2 (26.0–38.7)
	GT	140 (110–155)	85 (80–90)	33.9 (27.3–40.0)
	GG	135 (106–150)	90 (80–101)	33.0 (26.7–34.3)
**276 G/T** **(rs1501299)** ** *ADIPOQ* **	GG	130 (110–145)	85 (80–95)	33.9 (29.1–36.3)
	GT	130 (110–140)	90 (80–95)	33.6 (26.7–41.5)
	TT	130	80	33.9
** *+* ** **102 T/G** **(rs2275737)** ** *ADIPOR1* **	TT	130.0 (120–150)	80 (80–85)	31.9 (27.8–34.3)
	TG	130 (110–140)	85 (80–90)	34.1 (28.8–37.7)
	GG	140 (116–160)	90 (87.5–100.5)	36.5 (28.4–40.9)
**–106T/C** **(rs2275738)** ** *ADIPOR1* **	TT	140 (105–160)	90 (80–101)	34.3 (26.7–40.0)
	CT	130 (110–140)	85 (80–90)	34.7 (31.8–41.5)
	CC	130 (120–160)	80 (80–100)	28.3 (26.0–33.9)
**+219 A/T** **(rs11061971)** ** *ADIPOR2* **	AA	125 (115–140)	87.5 (75–101)	34.3 (26.7–40.0)
	AT	130 (130–155)	80 (80–90)	31.2 (26–36.6)
	TT	135 (123–140)	87.5 (80–90)	34.2 (30.9–40.8)
**+795 G/A** **(rs16928751)** ** *ADIPOR2* **	GG	130 (110–140)	**82.5 (70–90) ***	32 (26–34.6)
	GA	130 (110–130)	85 (80–104)	38.7 (33.5–41.5) *
	AA	140 (110–140)	90 (75–90)	30 (23–30.0)
**+2548 G/A** **(rs7799039)** ** *LEP* **	GG	**157.5 (140.0–170.0) ***	85.0 (80.0–90.0)	33.3 (27.3–34.7)
	GA	130.0 (113.0–140.0)	80.0 (80.0–95.5)	33.2 (28.3–40.0)
	AA	130.0 (110.0–140.0)	90.0 (70.0–90.0)	33.9 (26.0–40.8)
**–174G/C** **(rs1800795)** ** *IL6* **	CC	130 (120.0–147.5)	80 (75.0–90.0)	33.6 (32.3–35.1)
	GG	140 (121.3–148.8)	80 (80.0–90.0)	33.9 (25.3–33.9)

Notes. Me: median; Q1–Q3: the first and third quartiles; BP systolic, systolic blood pressure; BP diastolic, diastolic blood pressure; BMI, body mass index. * *p* value < 0.05 as compared to control (highlighted in bold).

## Data Availability

The relevant data and its supplemental data can be found in the article or obtained from the corresponding author upon request.

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
