# Peer review of "The Polymorphism of Metabolic and Immune Mechanisms Controlling Genes in Type 2 Diabetes Mellitus"

_genes, 2025, doi:10.3390/genes16091116_

Round 1

Reviewer 1 Report

Comments and Suggestions for Authors

The manuscript investigates the association of several SNPs in IL6, leptin, ADIPOQ, ADIPOR1, and ADIPOR2 genes with clinical features of metabolic syndrome and type 2 diabetes mellitus in a Crimean population. The study addresses a relevant and timely topic in the field of genetic epidemiology and metabolic disorders. The introduction gives adequate background and the methods are described in detail. The iThenticate score is OK and the topic is within the scope of the Geners journal. However, I have also some comments and suggestions for the authors:

Detailed comments:

The abstract is overly long and contains excessive methodological details. Consider condensing and focusing on the key findings and their implications.

Line 49, it should also be stated that drugs and other chemical substances may induce DM, i.e. please cite https://doi.org/10.56782/pps.183

Line 97, here, the aim of the study should be clearly stated

Table 1, I assume that “Me” stands for median, but it should be clearly stated in the table caption

Table 1, the authors should add some standard recommended values for the parameters (excluding age, of course)

Table 2 is overloaded and difficult to read. Consider splitting by gene or highlighting statistically significant results in bold.

Line 102, it should be either “Ninety-four” or (preferably) “94” but not “Ninety-fore”

Lines 235-250: The discussion repeats background information instead of focusing on how the present results support or contradict prior studies.

Line 286, the limitations of this study should be clearly stated, i.e. small sample size, single population

Conclusions: The recommendations for future research are generic. Suggest including specific methodological improvements (i.e., larger sample size, inclusion of GWAS-validated SNPs, longitudinal design).

Author Response

Comment 1.The abstract is overly long and contains excessive methodological details. Consider condensing and focusing on the key findings and their implications.

Response 1. We are agree with the comment, therefore we shortened and revised the abstract.

Comment 2.Line 49, it should also be stated that drugs and other chemical substances may induce DM, i.e. please cite https://doi.org/10.56782/pps.183-

Response 2. We agree with the comment, and therefore added the information and reference.

Comment 3. Line 97, here, the aim of the study should be clearly stated.

Response 3.We agree with the comment, and therefore specified the aim of the study.

Comment 4. Table 1, I assume that “Me” stands for median, but it should be clearly stated in the table caption.

Response 4.We agree with the comment, therefore all abbreviations and their meanings have been added to the legend of the tables. .

Comment 5. Table 1, the authors should add some standard recommended values for the parameters (excluding age, of course)

Response 5.We agree with the comment, therefore standard recommended values for the parameters were added to the table 1.

Comment 6. Table 2 is overloaded and difficult to read. Consider splitting by gene or highlighting statistically significant results in bold

Also, we have improved English to more clearly express the research.

Response 6. We agree with the comment, therefore,Table 2 is divided onto two tables-â„–â„–2 and 3 , besides that, statistically significant results were highlighted in bold.

Comment 7. Line 102, it should be either “Ninety-four” or (preferably) “94” but not “Ninety-fore”

Response 7. We agree with the comment, therefore, correction onto “Ninety-four” was made.

Comment 8. Lines 235-250: The discussion repeats background information instead of focusing on how the present results support or contradict prior studies.

Response 8. We agree with the comment, therefore, we have revised discussion part to the way of highlighting our research in supporting / contradicting prior studies, as well as further line of investigations in the given field.

Comment 9. Line 286, the limitations of this study should be clearly stated, i.e. small sample size, single population

Response 9.  We agree with the comment, therefore, we added recommended information about limitations of the study.

Comment 10. The recommendations for future research are generic. Suggest including specific methodological improvements (i.e., larger sample size, inclusion of GWAS-validated SNPs, longitudinal design

Response 10. We agree with the comment, therefore, we specified recommendations.

Reviewer 2 Report

Comments and Suggestions for Authors

This is an interesting study concerning the possible correlation among polymorphisms of metabolic and immune mechanisms controlling genes in type 2 diabetes mellitus. This is interesting, as there is a lack of research on the interactions between other genes associated with PPAR-γ and their role 92 in the pathogenesis of these diseases.

Authors shows expertise in the field. Ethical criteria are satisfied. Methods. The use of genome-wide association study (GWAS)  and genetic risk scores (GRS) are correct. The statistical package is also appropriate.

Minor points that should be addressed:

  • be careful about the format and repaired when necessary (i.e. line 50) and other paragraph´s initiations.
  • The exclusion criteria “Unstable T2DM (target glycated hemoglobin level (HbA1c) >7.0%; target fasting plasma glucose level >7.0 mmol/L (2 h after meals, >9.0 mmol/L)” should be justified. It is possible that these patients reinforce the correlation with the studied polymorphisms.
  • A statement about the C-174G variant of the IL-6 gene is included at the abstract and the final paragraph of discussion. Final statements about the other genes studies (IL-6, leptin, ADIPOQ, ADIPOR-1, ADIPOR-2 according to the abstract and the introduction) would be also included. In addition, other candidate genes could be suggested for future studies. AP-1, NF-kB or FFA. These suggestions would be introduced at the final part of discussion.
  • Conclusion is correct, but too general- The current conclusion it is as expected. Authors should include some particular conclusion directly related from the results and genes studies. As suggestion, the finding that the presence of the C-174G variant of the IL-6 increases the risk of MS when the CC genotype is present would be included at the final conclusion (translation of lines 278-281 to the referred section).
  • Line 183: Table 1 or 2? Please, check it and confirm Table number. Parameters such as IL-6 concentrations are not found in Table 1.

Author Response

Comment 1.Be careful about the format and repaired when necessary (i.e. line 50) and other paragraph´s initiations.

Response 1.Thank you for the  pointing this out, we repaired paragraphs' format.

Comment 2. The exclusion criteria “Unstable T2DM (target glycated hemoglobin level (HbA1c) >7.0%; target fasting plasma glucose level >7.0 mmol/L (2 h after meals, >9.0 mmol/L)” should be justified. It is possible that these patients reinforce the correlation with the studied polymorphisms.

Response 2. We agree with the comment, therefore we changed "Unstable T2DM" on “A severe course of T2DM”, according to given cited article. This group was excluded because our aim was to reveal risk in MS, which is NOT associated with the severe course of T2DM.

Comment 3. A statement about the C-174G variant of the IL-6 gene is included at the abstract and the final paragraph of discussion. Final statements about the other genes studies (IL-6, leptin, ADIPOQ, ADIPOR-1, ADIPOR-2 according to the abstract and the introduction) would be also included. In addition, other candidate genes could be suggested for future studies. AP-1, NF-kB or FFA. These suggestions would be introduced at the final part of discussion.

Response 3. We agree with the comment, therefore mentioned suggestion has  moved to recommended part with appropriate correction.

Comment 4. Conclusion is correct, but too general- The current conclusion it is as expected. Authors should include some particular conclusion directly related from the results and genes studies. As suggestion, the finding that the presence of the C-174G variant of the IL-6 increases the risk of MS when the CC genotype is present would be included at the final conclusion (translation of lines 278-281 to the referred section)

Response 4. We agree with the comment, therefore we specified conclusion according the given recommendations.

Comment 5. Line 183: Table 1 or 2? Please, check it and confirm Table number. Parameters such as IL-6 concentrations are not found in Table 1.

Response 5. Thank you for the  pointing this out, we corrected the line on "Table 2,3".

Round 2

Reviewer 1 Report

Comments and Suggestions for Authors

The Authors have revised and improved their work. Current version can be accepted.